# Wasserstein Weisfeiler-Lehman Graph Kernels

**Matteo Togninalli**[1,2,*]
matteo.togninalli@bsse.ethz.ch

**Elisabetta Ghisu**[1,2,*]
elisabetta.ghisu@bsse.ethz.ch

**Felipe Llinares-López**[1,2]
felipe.llinares@bsse.ethz.ch

**Bastian Rieck**[1,2]
bastian.rieck@bsse.ethz.ch

**Karsten Borgwardt**[1,2]
karsten.borgwardt@bsse.ethz.ch

[1]DEPARTMENT OF BIOSYSTEMS SCIENCE AND ENGINEERING, ETH ZURICH, SWITZERLAND
[2]SIB SWISS INSTITUTE OF BIOINFORMATICS, SWITZERLAND
[*]These authors contributed equally

## Abstract

Most graph kernels are an instance of the class of $\mathcal{R}$-Convolution kernels, which measure the similarity of objects by comparing their substructures. Despite their empirical success, most graph kernels use a naive aggregation of the final set of substructures, usually a sum or average, thereby potentially discarding valuable information about the distribution of individual components. Furthermore, only a limited instance of these approaches can be extended to continuously attributed graphs. We propose a novel method that relies on the Wasserstein distance between the node feature vector distributions of two graphs, which allows finding subtler differences in data sets by considering graphs as high-dimensional objects rather than simple means. We further propose a Weisfeiler–Lehman-inspired embedding scheme for graphs with continuous node attributes and weighted edges, enhance it with the computed Wasserstein distance, and thereby improve the state-of-the-art prediction performance on several graph classification tasks.

## 1 Introduction

Graph-structured data have become ubiquitous across domains over the last decades, with examples ranging from social and sensor networks to chemo- and bioinformatics. Graph kernels [45] have been highly successful in dealing with the complexity of graphs and have shown good predictive performance on a variety of classification problems [27, 38, 47]. Most graph kernels rely on the $\mathcal{R}$-Convolution framework [18], which decomposes structured objects into substructures to compute local similarities that are then aggregated. Although being successful in several applications, $\mathcal{R}$-Convolution kernels on graphs have limitations: (1) the simplicity of the way in which the similarities between substructures are aggregated might limit their ability to capture complex characteristics of the graph; (2) most proposed variants do not generalise to graphs with high-dimensional continuous node attributes, and extensions are far from being straightforward. Various solutions have been proposed to address point (1). For example, Fröhlich et al. [15] introduced kernels based on the optimal assignment of node labels for molecular graphs, although these kernels are not positive definite [43]. Recently, another approach was proposed by Kriege et al. [25], which employs a Weisfeiler–Lehman based colour refinement scheme and uses an optimal assignment of the nodes to compute the kernel. However, this method cannot handle continuous node attributes, leaving point (2) as an open problem.

To overcome both limitations, we propose a method that combines the most successful vectorial graph representations derived from the graph kernel literature with ideas from optimal transport theory, which have recently gained considerable attention. In particular, improvements of the computational strategies to efficiently obtain Wasserstein distances [1, 8] have led to many applications in machine learning that use it for various purposes, ranging from generative models [2] to new loss functions [14]. In graph applications, notions from optimal transport were used to tackle the graph alignment problem [46]. In this paper, we provide the theoretical foundations of our method, define a new graph kernel formulation, and present successful experimental results. Specifically, our main contributions can be summarised as follows:

- We present the graph Wasserstein distance, a new distance between graphs based on their node feature representations, and we discuss how kernels can be derived from it.
- We introduce a Weisfeiler–Lehman-inspired embedding scheme that works for both categorically labelled and continuously attributed graphs, and we couple it with our graph Wasserstein distance;
- We outperform the state of the art for graph kernels on traditional graph classification benchmarks with continuous attributes.

## 2 Background: graph kernels and Wasserstein distance

In this section, we introduce the notation that will be used throughout the manuscript. Moreover, we provide the necessary background on graph kernel methods and the Wasserstein distance.

### 2.1 Graph kernels

Kernels are a class of similarity functions that present attractive properties to be used in learning algorithms [36]. Let $\mathcal{X}$ be a set and $k\colon \mathcal{X} \times \mathcal{X} \to \mathbb{R}$ be a function associated with a Hilbert space $\mathcal{H}$, such that there exists a map $\phi\colon \mathcal{X} \to \mathcal{H}$ with $k(x, y) = \langle \phi(x), \phi(y) \rangle_{\mathcal{H}}$. Then, $\mathcal{H}$ is a reproducing kernel Hilbert space (RKHS) and $k$ is said to be a positive definite kernel. A positive definite kernel can be interpreted as a dot product in a high-dimensional space, thereby permitting its use in any learning algorithm that relies on dot products, such as support vector machines (SVMs), by virtue of the *kernel trick* [35]. Because ensuring positive definiteness is not always feasible, many learning algorithms were recently proposed to extend SVMs to indefinite kernels [3, 26, 29, 30].

We define a graph as a tuple $G = (V, E)$, where $V$ and $E$ denote the set of nodes and edges, respectively; we further assume that the edges are undirected. Moreover, we denote the cardinality of nodes and edges for $G$ as $|V| = n_G$ and $|E| = m_G$. For a node $v \in V$, we write $\mathcal{N}(v) = \{u \in V \mid (v, u) \in E\}$ and $|\mathcal{N}(v)| = \deg(v)$ to denote its first-order neighbourhood. We say that a graph is *labelled* if its nodes have categorical labels. A label on the nodes is a function $l\colon V \to \Sigma$ that assigns to each node $v$ in $G$ a value $l(v)$ from a finite label alphabet $\Sigma$. Additionally, we say that a graph is *attributed* if for each node $v \in V$ there exists an associated vector $a(v) \in \mathbb{R}^m$. In this paper, $a(v)$ are the node attributes and $l(v)$ are the categorical node labels of node $v$. In particular, the node attributes are high-dimensional continuous vectors, whereas the categorical node labels are assumed to be integer numbers (encoding either an ordered discrete value or a category). With the term "node labels", we will implicitly refer to categorical node labels. Finally, a graph can have weighted edges, and the function $w\colon E \to \mathbb{R}$ defines the weight $w(e)$ of an edge $e := (v, u) \in E$.

Kernels on graphs are generally defined using the $\mathcal{R}$-Convolution framework by [18]. The main idea is to decompose graph $G$ into substructures and to define a kernel value $k(G, G')$ as a combination of substructure similarities. A pioneer kernel on graphs was presented by [19], where node and edge attributes are exploited for label sequence generation using a random walk scheme. Successively, a more efficient approach based on shortest paths [5] was proposed, which computes each kernel value $k(G, G')$ as a sum of the similarities between each shortest path in $G$ and each shortest path in $G'$. Despite the practical success of $\mathcal{R}$-Convolution kernels, they often rely on aggregation strategies that ignore valuable information, such as the distribution of the substructures. An example is the Weisfeiler–Lehman (WL) subtree kernel or one of its variants [33, 37, 38], which generates graph-level features by summing the contribution of the node representations. To avoid these simplifications, we want to use concepts from optimal transport theory, such as the Wasserstein distance, which can help to better capture the similarities between graphs.

## 2.2 Wasserstein distance

The Wasserstein distance is a distance function between probability distributions defined on a given metric space. Let $\sigma$ and $\mu$ be two probability distributions on a metric space $M$ equipped with a ground distance $d$, such as the Euclidean distance.

**Definition 1.** *The $L^p$-Wasserstein distance for $p \in [1, \infty)$ is defined as*

$$W_p(\sigma, \mu) := \left( \inf_{\gamma \in \Gamma(\sigma, \mu)} \int_{M \times M} d(x, y)^p \, \mathrm{d}\gamma(x, y) \right)^{\frac{1}{p}}, \tag{1}$$

*where $\Gamma(\sigma, \mu)$ is the set of all transportation plans $\gamma \in \Gamma(\sigma, \mu)$ over $M \times M$ with marginals $\sigma$ and $\mu$ on the first and second factors, respectively.*

The Wasserstein distance satisfies the axioms of a metric, provided that $d$ is a metric (see the monograph of Villani [44], chapter 6, for a proof). Throughout the paper, we will focus on the distance for $p = 1$ and we will refer to the $L^1$-Wasserstein distance when mentioning the Wasserstein distance, unless noted otherwise.

The Wasserstein distance is linked to the optimal transport problem [44], where the aim is to find the most "inexpensive" way, in terms of the ground distance, to transport all the probability mass from distribution $\sigma$ to match distribution $\mu$. An intuitive illustration can be made for the 1-dimensional case, where the two probability distributions can be imagined as piles of dirt or sand. The Wasserstein distance, sometimes also referred to as the earth mover's distance [34], can be interpreted as the minimum effort required to move the content of the first pile to reproduce the second pile.

In this paper, we deal with finite sets of node embeddings and not with continuous probability distributions. Therefore, we can reformulate the Wasserstein distance as a sum rather than an integral, and use the matrix notation commonly encountered in the optimal transport literature [34] to represent the transportation plan. Given two sets of vectors $X \in \mathbb{R}^{n \times m}$ and $X' \in \mathbb{R}^{n' \times m}$, we can equivalently define the Wasserstein distance between them as

$$W_1(X, X') := \min_{P \in \Gamma(X, X')} \langle P, M \rangle. \tag{2}$$

Here, $M$ is the distance matrix containing the distances $d(x, x')$ between each element $x$ of $X$ and $x'$ of $X'$, $P \in \Gamma$ is a transport matrix (or joint probability), and $\langle \cdot, \cdot \rangle$ is the Frobenius dot product. The transport matrix $P$ contains the fractions that indicate how to transport the values from $X$ to $X'$ with the minimal total transport effort. Because we assume that the total mass to be transported equals 1 and is evenly distributed across the elements of $X$ and $X'$, the row and column values of $P$ must sum up to $1/n$ and $1/n'$, respectively.

## 3 Wasserstein distance on graphs

The unsatisfactory nature of the aggregation step of current $\mathcal{R}$-Convolution graph kernels, which may mask important substructure differences by averaging, motivated us to have a finer distance measure between structures and their components. In parallel, recent advances in optimisation solutions for faster computation of the optimal transport problem inspired us to consider this framework for the problem of graph classification. Our method relies on the following steps: (1) transform each graph into a set of node embeddings, (2) measure the Wasserstein distance between each pair of graphs, and (3) compute a similarity matrix to be used in the learning algorithm. Figure 1 illustrates the first two steps, and Algorithm 1 summarises the whole procedure. We start by defining an embedding scheme and illustrate how we integrate embeddings in the Wasserstein distance.

**Definition 2** (Graph Embedding Scheme). *Given a graph $G = (V, E)$, a graph embedding scheme $f \colon G \to \mathbb{R}^{|V| \times m}$, $f(G) = X_G$ is a function that outputs a fixed-size vectorial representation for each node in the graph. For each $v_i \in V$, the $i$-th row of $X_G$ is called the node embedding of $v_i$.*

Note that Definition 2 permits treating node labels, which are categorical attributes, as one-dimensional attributes with $m = 1$.

**Definition 3** (Graph Wasserstein Distance). *Given two graphs $G = (V, E)$ and $G' = (V', E')$, a graph embedding scheme $f \colon G \to \mathbb{R}^{|V| \times m}$ and a ground distance $d \colon \mathbb{R}^m \times \mathbb{R}^m \to \mathbb{R}$, we define the Graph Wasserstein Distance (GWD) as*

$$D_W^f(G, G') := W_1(f(G), f(G')). \tag{3}$$

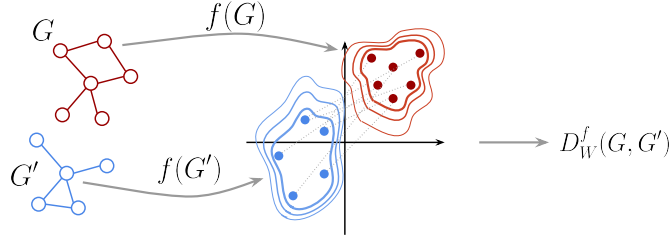

Figure 1: Visual summary of the graph Wasserstein distance. First, $f$ generates embeddings for two input graphs $G$ and $G'$. Then, the Wasserstein distance between the embedding distributions is computed.

We will now propose a graph embedding scheme inspired by the WL kernel on categorically labeled graphs, extend it to continuously attributed graphs with weighted edges, and show how to integrate it with the GWD presented in Definition 3.

## 3.1  Generating node embeddings

**The Weisfeiler–Lehman scheme.** The Weisfeiler–Lehman subtree kernel [37, 38], designed for labelled non-attributed graphs, looks at similarities among subtree patterns, defined by a propagation scheme on the graphs that iteratively compares labels on the nodes and their neighbours. This is achieved by creating a sequence of ordered strings through the aggregation of the labels of a node and its neighbours; those strings are subsequently *hashed* to create updated *compressed* node labels. With increasing iterations of the algorithm, these labels represent increasingly larger neighbourhoods of each node, allowing to compare more extended substructures.

Specifically, consider a graph $G = (V, E)$, let $\ell^0(v) = \ell(v)$ be the initial node label of $v$ for each $v \in V$, and let $H$ be the number of WL iterations. Then, we can define a recursive scheme to compute $\ell^h(v)$ for $h = 1, \ldots, H$ by looking at the ordered set of neighbours labels $\mathcal{N}^h(v) = \{\ell^h(u_0), \ldots, \ell^h(u_{\deg(v)-1})\}$ as

$$\ell^{h+1}(v) = \mathrm{hash}(\ell^h(v), \mathcal{N}^h(v)). \tag{4}$$

We call this procedure the WL labelling scheme. As in the original publication [37], we use perfect hashing for the hash function, so nodes at iteration $h + 1$ will have the same label if and only if their label and those of their neighbours are identical at iteration $h$.

**Extension to continuous attributes.** For graphs with continuous attributes $a(v) \in \mathbb{R}^m$, we need to improve the WL refinement step, whose original definition prohibits handling the continuous case. The key idea is to create an explicit propagation scheme that leverages and updates the current node features by averaging over the neighbourhoods. Although similar approaches have been implicitly investigated for computing node-level kernel similarities [27, 28], they rely on additional hashing steps for the continuous features. Moreover, we can easily account for edge weights by considering them in the average calculation of each neighbourhood. Suppose we have a continuous attribute $a^0(v) = a(v)$ for each node $v \in G$. Then, we recursively define

$$a^{h+1}(v) = \frac{1}{2}\left(a^h(v) + \frac{1}{\deg(v)} \sum_{u \in \mathcal{N}(v)} w\left((v, u)\right) \cdot a^h(u)\right). \tag{5}$$

When edge weights are not available, we set $w(u, v) = 1$. We consider the weighted average of the neighbourhood attribute values instead of a sum and add the $1/2$ factor because we want to ensure a similar scale of the features across iterations; in fact, we concatenate such features for building our proposed kernel (see Definition 4 for more details) and observe better empirical results with similarly scaled features. Although this is not a test of graph isomorphism, this refinement step can be seen as an intuitive extension for continuous attributes of the one used by the WL subtree kernel on categorical node labels, a widely successful baseline. Moreover, it resembles the propagation scheme used in many graph neural networks, which have proven to be successful for node classification on large data sets [9, 21, 22]. Finally, its ability to account for edge weights makes it applicable to all

types of graphs without having to perform a hashing step [27]. Further extensions of the refinement step to account for high-dimensional edge attributes are left for future work. A straightforward example would be to also apply the scheme on the *dual* graph (where each edge is represented as a node, and connectivity is established if two edges in the primal graph share the same node) to then combine the obtained kernel with the kernel obtained on primal graphs via appropriate weighting.

**Graph embedding scheme.** Using the recursive procedure described above, we propose a WL-based graph embedding scheme that generates node embeddings from the node labels or attributes of the graphs. In the following, we use $m$ to denote the dimensionality of the node attributes ($m = 1$ for the categorical labels).

**Definition 4** (WL features). *Let $G = (V, E)$ and let $H$ be the number of WL iterations. Then, for every $h \in \{0, \ldots, H\}$, we define the WL features as*

$$X_G^h = [x^h(v_1), \ldots, x^h(v_{n_G})]^T, \tag{6}$$

*where $x^h(\cdot) = \ell^h(\cdot)$ for categorically labelled graphs and $x^h(\cdot) = a^h(\cdot)$ for continuously attributed graphs. We refer to $X_G^h \in \mathbb{R}^{n_G \times m}$ as the* node features *of graph $G$ at iteration $h$. Then, the node embeddings of graph $G$ at iteration $H$ are defined as*

$$\begin{aligned} f^H \colon G &\to \mathbb{R}^{n_G \times (m(H+1))} \\ G &\mapsto \text{concatenate}(X_G^0, \ldots, X_G^H). \end{aligned} \tag{7}$$

We observe that a graph can be both *categorically labelled* and *continuously attributed*, and one could extend the above scheme by jointly considering this information (for instance, by concatenating the node features). However, we will leave this scenario as an extension for future work; thereby, we avoid having to define an appropriate distance measure between categorical and continuous data, as this is a long-standing issue [40].

## 3.2 Computing the Wasserstein distance

Once the node embeddings are generated by the graph embedding scheme, we evaluate the pairwise Wasserstein distance between graphs. We start by computing the ground distances between each pair of nodes. For categorical node features, we use the normalised Hamming distance:

$$d_{\text{Ham}}(v, v') = \frac{1}{H+1} \sum_{i=1}^{H+1} \rho(v_i, v_i'), \quad \rho(x, y) = \begin{cases} 1, & x \neq y \\ 0, & x = y \end{cases} \tag{8}$$

The Hamming distance can be pictured as the normalised sum of discrete metric $\rho$ on each of the features. The Hamming distance equals 1 when two vectors have no features in common and 0 when the vectors are identical. We use the Hamming distance as, in this case, the Weisfeiler–Lehman features are indeed categorical, and values carry no meaning. For continuous node features, on the other hand, we employ the Euclidean distance:

$$d_E(v, v') = ||v - v'||_2. \tag{9}$$

Next, we substitute the ground distance into the equation of Definition 1 and compute the Wasserstein distance using a network simplex method [31].

**Computational complexity.** Naively, the computation of the Wasserstein Distance has a complexity of $\mathcal{O}(n^3 log(n))$, with $n$ being the cardinality of the indexed set of node embeddings, i.e., the number of nodes in the two graphs. Nevertheless, efficient speedup tricks can be employed. In particular, approximations relying on Sinkhorn regularisation have been proposed [8], some of which reduce the computational burden to *near-linear time* while preserving accuracy [1]. Such speedup strategies become incredibly useful for larger data sets, i.e., graphs with thousands of nodes, and can be easily integrated into our method. See Appendix A.7 for a practical discussion.

## 4 From Wasserstein distance to kernels

From the graph Wasserstein distance, one can construct a similarity measure to be used in a learning algorithm. In this section, we propose a new graph kernel, state some claims about its (in)definiteness, and elaborate on how to use it for classifying graphs with continuous and categorical node labels.

---
**Algorithm 1** Compute Wasserstein graph kernel
---
**Input:** Two graphs $G_1$, $G_2$; graph embedding scheme $f^H$; ground distance $d$; $\lambda$.
**Output:** kernel value $k_{WWL}(G_1, G_2)$.
$X_{G_1} \leftarrow f^H(G_1)$; $X_{G_2} \leftarrow f^H(G_2)$ // Generate node embeddings
$D \leftarrow \text{pairwise\_dist}(X_{G_1}, X_{G_2}, d)$ // Compute the ground distance between each pair of nodes
$D_W(G_1, G_2) = \min_{P \in \Gamma} \langle P, D \rangle$ // Compute the Wasserstein distance
$k_W(G_1, G_2) \leftarrow e^{-\lambda D_W(G_1, G_2)}$
---

Table 1: Classification accuracies on graphs with categorical node labels. Comparison of Weisfeiler–Lehman kernel (WL), optimal assignment kernel (WL-OA), and our method (WWL).

| METHOD | MUTAG | PTC-MR | NCI1 | PROTEINS | D&D | ENZYMES |
|---|---|---|---|---|---|---|
| V | 85.39±0.73 | 58.35±0.20 | 64.22±0.11 | 72.12±0.19 | 78.24±0.28 | 22.72±0.56 |
| E | 84.17±1.44 | 55.82±0.00 | 63.57±0.12 | 72.18±0.42 | 75.49±0.21 | 21.87±0.64 |
| WL | 85.78±0.83 | 61.21±2.28 | 85.83±0.09 | 74.99±0.28 | 78.29±0.30 | 53.33±0.93 |
| WL-OA | **87.15±1.82** | 60.58±1.35 | **86.08±0.27** | **76.37±0.30**$^*$ | 79.15±0.33 | **58.97±0.82** |
| WWL | **87.27±1.50** | **66.31±1.21**$^*$ | 85.75±0.25 | 74.28±0.56 | **79.69±0.50** | **59.13±0.80** |

**Definition 5** (Wasserstein Weisfeiler–Lehman). *Given a set of graphs $\mathcal{G} = \{G_1, \ldots, G_N\}$ and the GWD defined for each pair of graph on their WL embeddings, we define the Wasserstein Weisfeiler–Lehman (WWL) kernel as*

$$K_{\text{WWL}} = e^{-\lambda D_W^{f_{\text{WL}}}}. \tag{10}$$

This is an instance of a Laplacian kernel, which was shown to offer favourable conditions for positive definiteness in the case of non-Euclidean distances [11]. Obtaining the WWL kernel concludes the procedure described in Algorithm 1. In the remainder of this section, we distinguish between the categorical WWL kernel, obtained on graphs with categorical labels, and the continuous WWL kernel, obtained on continuously attributed graphs via the graph embedding schemes described in Section 3.1.

For Euclidean spaces, obtaining positive definite kernels from distance functions is a well-studied topic [17]. However, the Wasserstein distance in its general form is not isometric, i.e., there is no metric-preserving mapping to an $L^2$-norm, as the metric space it induces strongly depends on the chosen ground distance [12]. Therefore, despite being a metric, it is not necessarily possible to derive a positive definite kernel from the Wasserstein distance in its general formulation, because the classical approaches [17] cannot be applied here. Nevertheless, as a consequence of using the Laplacian kernel [11], we can show that, in the setting of categorical node labels, the obtained kernel is positive definite.

**Theorem 1.** *The categorical WWL kernel is positive definite for all $\lambda > 0$.*

For a proof, see Sections A.1 and A.1.1 in the Appendix. By contrast, for the continuous case, establishing the definiteness of the obtained kernel remains an open problem. We refer the reader to Section A.1.2 in the supplementary materials for further discussions and conjectures.

Therefore, to ensure the theoretical and practical correctness of our results *in the continuous case*, we employ recently developed methods for learning with indefinite kernels. Specifically, we use learning methods for Kreĭn spaces, which have been specifically designed to work with indefinite kernels [30]; in general, kernels that are not positive definite induce reproducing kernel Kreĭn spaces (RKKS). These spaces can be seen as a generalisation of reproducing kernel Hilbert spaces, with which they share similar mathematical properties, making them amenable to machine learning techniques. Recent algorithms [26, 29] are capable of solving learning problems in RKKS; their results indicate that there are clear benefits (in terms of classification performance, for example) of learning in such spaces. Therefore, when evaluating WWL, we will use a Kreĭn SVM (KSVM, [26]) as a classifier for the case of continuous attributes.

Table 2: Classification accuracies on graphs with continuous node and/or edge attributes. Comparison of hash graph kernel (HGK-WL, HGK-SP), GraphHopper kernel (GH), and our method (WWL).

| METHOD | ENZYMES | PROTEINS | IMDB-B | BZR | COX2 | BZR-MD | COX2-MD |
|---|---|---|---|---|---|---|---|
| VH-C | 47.15±0.79 | 60.79±0.12 | 71.64±0.49 | 74.82±2.13 | 48.51±0.63 | 66.58±0.97 | 64.89±1.06 |
| RBF-WL | 68.43±1.47 | 75.43±0.28 | 72.06±0.34 | 80.96±1.67 | 75.45±1.53 | **69.13±1.27** | 71.83±1.61 |
| HGK-WL | 63.04±0.65 | 75.93±0.17 | 73.12±0.40 | 78.59±0.63 | **78.13±0.45** | 68.94±0.65 | 74.61±1.74 |
| HGK-SP | 66.36±0.37 | 75.78±0.17 | 73.06±0.27 | 76.42±0.72 | 72.57±1.18 | 66.17±1.05 | 68.52±1.00 |
| GH | 65.65±0.80 | 74.78±0.29 | 72.35±0.55 | 76.49±0.99 | 76.41±1.39 | **69.14±2.08** | 66.20±1.05 |
| WWL | **73.25±0.87**$^*$ | **77.91±0.80**$^*$ | **74.37±0.83**$^*$ | **84.42±2.03**$^*$ | **78.29±0.47** | **69.76±0.94** | **76.33±1.02** |

## 5 Experimental evaluation

In this section, we analyse how the performance of WWL compares with state-of-the-art graph kernels. In particular, we empirically observe that WWL (1) is competitive with the best graph kernel for categorically labelled data, and (2) outperforms all the state-of-the-art graph kernels for attributed graphs.

### 5.1 Data sets

We report results on real-world data sets from multiple sources [6, 38, 45] and use either their continuous attributes or categorical labels for evaluation. In particular, MUTAG, PTC-MR, NCI1, and D&D are equipped with categorical node labels only; ENZYMES and PROTEINS have both categorical labels and continuous attributes; IMDB-B, BZR, and COX2 only contain continuous attributes; finally, BZR-MD and COX2-MD have both continuous node attributes and edge weights. Further information on the data sets is available in Supplementary Table A.1. Additionally, we report results on synthetic data (SYNTHIE and SYNTHETIC-NEW) in Appendix A.5. All the data sets have been downloaded from Kersting et al. [20].

### 5.2 Experimental setup

We compare WWL with state-of-the-art graph kernel methods from the literature and relevant baselines, which we trained ourselves on the same splits (see below). In particular, for the categorical case, we compare with WL [37] and WL-OA [25] as well as with the vertex (V) and edge (E) histograms. Because [25] already showed that the WL-OA is superior to previous approaches, we do not include the whole set of kernels in our comparison. For the continuously attributed data sets, we compare with two instances of the hash graph kernel (HGK-SP; HGK-WL) [27] and with the GraphHopper (GH) [10]. For comparison, we additionally use a continuous vertex histogram (VH-C), which is defined as a radial basis function (RBF) kernel between the sum of the graph node embeddings. Furthermore, to highlight the benefits of using the Wasserstein distance in our method, we replace it with an RBF kernel. Specifically, given two graphs $G_1 = (V_1, E_1)$ and $G_2 = (V_2, E_2)$, with $|V_1| = n_1$ and $|V_2| = n_2$, we first compute the Gaussian kernel between each pair of the node embeddings obtained in the same fashion as for WWL; therefore, we obtain a kernel matrix between node embeddings $K' \in n_1 \times n_2$. Next, we sum up the values $K_s = \sum_{i=1}^{n_1} \sum_{j=1}^{n_2} K'_{i,j}$ and set $K(G_1, G_2) = K_s$. This procedure is repeated for each pair of graphs to obtain the final graph kernel matrix. We refer to this baseline as RBF-WL.

As a classifier, we use an SVM (or a KSVM in the case of WWL) and 10-fold cross-validation, selecting the parameters on the training set only. We repeat each cross-validation split 10 times and report the average accuracy. We employ the same split for each evaluated method, thereby guaranteeing a fully comparable setup among all evaluated methods. Please refer to Appendix A.6 for details on the hyperparameter selection.

**Implementation and computing infrastructure** Available Python implementations can be used to compute the WL kernel [41] and the Wasserstein distance [13]. We leverage these resources and

make our code publicly available[1]. We use the original implementations provided by the respective authors to compute the WL-OA, HGK, and GH methods. All our analyses were performed on a shared server running Ubuntu 14.04.5 LTS, with 4 CPUs (Intel Xeon E7-4860 v2 @ 2.60GHz) each with 12 cores and 24 threads, and 512 GB of RAM.

## 5.3 Results and discussion

The results are evaluated by classification accuracy and summarised in Table 1 and Table 2 for the categorical labels and continuous attributes, respectively[2].

### 5.3.1 Categorical labels

On the categorical data sets, WWL is comparable to the WL-OA kernel; however, it improves over the classical WL. In particular, WWL largely improves over WL-OA in PTC-MR and is slightly better on D&D, whereas WL-OA is better on NCI1 and PROTEINS.

Unsurprisingly, our approach is comparable to the WL-OA, whose main idea is to solve the optimal assignment problem by defining Dirac kernels on histograms of node labels, using multiple iterations of WL. This formulation is similar to the one we provide for categorical data, but it relies on the optimal assignment rather than the optimal transport; therefore, it requires one-to-one mappings instead of continuous transport maps. Besides, we solve the optimal transport problem on the concatenated embeddings, hereby jointly exploiting representations at multiple WL iterations. Contrarily, the WL-OA performs an optimal assignment at each iteration of WL and only combines them in the second stage. However, the key advantage of WWL over WL-OA is its capacity to account for continuous attributes.

### 5.3.2 Continuous attributes

In this setting, WWL significantly outperforms the other methods on 4 out of 7 data sets, is better on another one, and is on a par on the remaining 2. We further compute the average rank of each method in the continuous setting, with WWL scoring as first. The ranks calculated from Table 2 are WWL = 1, HGK-WL = 2.86, RBF-WL = 3.29, HGK-SP = 4.14, and VH-C = 5.86. This is a remarkable improvement over the current state of the art, and it indeed establishes a new one. When looking at the average rank of the method, WWL always scores first. Therefore, we raise the bar in kernel graph classification for attributed graphs. As mentioned in Section 4, the kernel obtained from continuous attributes is not necessarily positive definite. However, we empirically observe the kernel matrices to be positive definite (up to a numerical error), further supporting our theoretical considerations (see Appendix A.1). In practice, the difference between the results obtained from classical SVMs in RKHS and the results obtained with the KSVM approach is negligible.

**Comparison with hash graph kernels** The hash graph kernel (HGK) approach is somewhat related to our propagation scheme. By using multiple hashing functions, the HGK method is capable of extending certain existing graph kernels to the continuous setting. This helps to avoid the limitations of perfect hashing, which cannot express small differences in continuous attributes. A drawback of the random hashing performed by HGK is that it requires additional parameters and introduces a stochastic element to the kernel matrix computation. By contrast, our propagation scheme is fully continuous and uses the Wasserstein distance to capture small differences in distributions of continuous node attributes. Moreover, the observed performance gap suggests that an entirely continuous representation of the graphs provides clear benefits over the hashing.

## 6 Conclusion

In this paper, we present a new family of graph kernels, the Wasserstein Weisfeiler–Lehman (WWL) graph kernels. Our experiments show that WWL graph kernels outperform the state of the art for

graph classification in the scenario of continuous node attributes, while matching the state of the art in the categorical setting. As a line of research for future work, we see great potential in the runtime improvement, thus, allowing applications of our method on regimes and data sets with larger graphs. In fact, preliminary experiments (see Section A.7 as well as Figure A.1 in the Appendix) already confirm the benefit of Sinkhorn regularisation when the average number of nodes in the graph increases. In parallel, it would be beneficial to derive approximations of the explicit feature representations in the RKKS, as this would also provide a consistent speedup. We further envision that major theoretical contributions could be made by defining theoretical bounds to ensure the positive definiteness of the WWL kernel in the case of continuous node attributes. Finally, optimisation objectives based on optimal transport could be employed to develop new algorithms based on graph neural networks [9, 21]. On a more general level, our proposed method provides a solid foundation of the use of optimal transport theory for kernel methods and highlights the large potential of optimal transport for machine learning.

**Acknowledgments**

This work was funded in part by the Horizon 2020 project CDS-QUAMRI, Grant No. 634541 (E.G., K.B.), the Alfried Krupp Prize for Young University Teachers of the Alfried Krupp von Bohlen und Halbach-Stiftung (B.R., K.B.), and the SNSF Starting Grant "Significant Pattern Mining" (F.L., K.B.).

## Footnotes

[1] https://github.com/BorgwardtLab/WWL

[2] The best performing methods up to the resolution implied by the standard deviation across repetitions are highlighted in boldface. Additionally, to evaluate significance we perform 2-sample $t$-tests with a significance threshold of $0.05$ and Bonferroni correction for multiple hypothesis testing within each data set, significantly outperforming methods are denoted by an asterisk.

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
