[Supplementary Material]

# A Appendix

## A.1 Extended considerations on WWL definiteness

We will now discuss the positive definite nature of our WWL kernel.

In general, whether distances obtained from optimal transport problems can be used to create positive definite kernels remains an open research question. Several attempts to draw general conclusions on the definiteness of the Wasserstein distance were unsuccessful, but insightful results on particular cases were obtained along the way. First, we collect some of these contributions and use them to prove that our WWL kernel for categorical embeddings is positive definite. Next, we elaborate further on the continuous embeddings case, for which we provide conjectures on practical conditions to obtain a positive definite kernel.

Before proceeding, let us reiterate some useful notions.

**Definition 6.** *[36] A symmetric function* $k\colon \mathcal{X} \times \mathcal{X} \to \mathbb{R}$ *is called a* positive definite (pd) kernel *if it satisfies the condition*

$$\sum_{i,j=1}^{n} c_i c_j K_{ij} \geq 0, \ \ with \ K_{ij} = k(x_i, x_j), \tag{11}$$

*for every* $c_i \in \mathbb{R}$, $n \in \mathbb{N}$ *and* $x_i \in \mathcal{X}$.

The matrix of kernel values $K$ with entries $K_{ij}$ is called the *Gram matrix* of $k$ with respect to $x_1, \ldots, x_n$. A *conditional* positive definite (cpd) kernel is a function that satisfies Equation 11 for all $c_i \in \mathbb{R}$ with $\sum_{i=1}^{n} c_i = 0$. By analogy, a conditional negative definite (cnd) kernel is a function that satisfies $\sum_{i,j=1}^{n} c_i c_j K_{ij} \leq 0$ for all $c_i \in \mathbb{R}$ with $\sum_{i=1}^{n} c_i = 0$.

For Euclidean spaces, obtaining kernels from distance functions is a well-studied topic.

**Proposition 1.** *[17] Let* $d(x, x')$ *be a symmetric, non-negative distance function with* $d(x, x) = 0$. *If* $d$ *is isometric to an* $L^2$-norm, then

$$k_d^{\mathrm{nd}}(x, x') = -d(x, x')^{\beta}, \ \ \beta \in [0, 2] \tag{12}$$

*is a valid cpd kernel.*

However, the Wasserstein distance in its general form is not isometric to an $L^2$-norm, as the metric space it induces strongly depends on the chosen ground distance [12]. Recently, Feragen et al. [11] argued that many types of data, including probability distributions, do not always reside in Euclidean spaces. Therefore, they define the family of exponential kernels relying on a non-Euclidean distance $d$ as

$$k(x, x') = e^{-\lambda d(x,x')^q} \ \ \text{for} \ \ \lambda, q > 0, \tag{13}$$

and, based on earlier considerations from Berg et al. [4], show that, under certain conditions, the Laplacian kernel ($q = 1$ in Equation 13) is positive definite.

**Proposition 2.** *[11] The geodesic Laplacian kernel is positive definite for all* $\lambda > 0$ *if and only if the geodesic distance* $d$ *is conditional negative definite.*

Once again, considerations on the negative definiteness of Wasserstein distance functions cannot be made on the general level. Certain ground distances, however, *guarantee* the negative definiteness of the resulting Wasserstein distance. In particular, the Wasserstein distance with the discrete metric (i.e., $\rho$ in Equation 8) as the ground distance was proved to be conditional negative definite [16].

We will now leverage these results to prove that the Wasserstein distance equipped with the Hamming ground distance is conditional negative definite; therefore, it yields positive definite kernels for the categorical WL embeddings.

### A.1.1 The case of categorical embeddings

When generating node embeddings using the Weisfeiler–Lehman labelling scheme with a shared dictionary across all the graphs, the solutions to the optimal transport problem are also shared across iterations. We denote the Weisfeiler–Lehman embedding scheme as defined in Definition 4 as $f_{\mathrm{WL}}^H$, and let $D_W^{f_{\mathrm{WL}}}$ be the corresponding GWD on a set of graphs $\mathcal{G}$ with categorical labels. Let $d_{\mathrm{Ham}}(v, v')$ of Equation 8 be the ground distance of $D_W^{f_{\mathrm{WL}}}$. Then, the following useful results hold.

**Lemma 1.** *If a transportation plan $\gamma$ with transport matrix $P$ is optimal in the sense of Definition 1 for distances $d_{\mathrm{Ham}}$ between embeddings obtained with $f_{WL}^H$, then it is also optimal for the discrete distances $d_{\mathrm{disc}}$ between the $H$-th iteration values obtained with the Weisfeiler–Lehman procedure.*

*Proof.* See Appendix A.2.

**Lemma 2.** *If a transportation plan $\gamma$ with transport matrix $P$ is optimal in the sense of Definition 1 for distances $d_{\mathrm{Ham}}$ between embeddings obtained with $f_{\mathrm{WL}}^H$, then it is also optimal for distances $d_{\mathrm{Ham}}$ between embeddings obtained with $f_{\mathrm{WL}}^{H-1}$.*

*Proof.* See Appendix A.3.

Therefore, we postulate that the Wasserstein distance between categorical WL node embeddings is a conditional negative definite function.

**Theorem 2.** $D_W^{f_{\mathrm{WL}}}(\cdot, \cdot)$ *is a conditional negative definite function.*

*Proof.* See Appendix A.4.

**Proof of Theorem 1.** Theorem 2 in light of Proposition 2 implies that the WWL kernel of Definition 5 is positive definite for all $\lambda > 0$. □

We will now consider the case of the definiteness of kernels in the continuous setting.

### A.1.2 The case of continuous embeddings

On one hand, in the categorical case, we proved the positive definiteness of our kernel. On the other hand, the continuous case is considerably harder to tackle. We conjecture that, under certain conditions, the same might hold for continuous features. Although we do not have a formal proof yet, in what follows, we discuss arguments to support this conjecture, which seems to agree with our empirical findings.[3]

The *curvature* of the metric space induced by the Wasserstein metric for a given ground distance plays an important role. First, we need to define *Alexandrov spaces*.

**Definition 7** (Alexandrov space)**.** *Given a metric space and a real number $k$, the space is called an Alexandrov space if its sectional curvature is $\geq k$.*

Roughly speaking, the curvature indicates to what extent a geodesic triangle will be deformed in the space. The case of $k = 0$ is special as no distortion is happening here—hence, spaces that satisfy this property are called *flat*. The concept of Alexandrov spaces is required in the following proposition, taken from a theorem by Feragen et al. [11], which shows the relationship between a kernel and its underlying metric space.

**Proposition 3.** *The geodesic Gaussian kernel (i.e., $q = 2$ in Equation 13) is positive definite for all $\lambda > 0$ if and only if the underlying metric space $(X, d)$ is flat in the sense of Alexandrov, i.e., if any geodesic triangle in $X$ can be isometrically embedded in a Euclidean space.*

However, it is unlikely that the space induced by the Wasserstein distance is locally flat, as not even the geodesics (i.e., a generalisation of the shortest path to arbitrary metric spaces) between graph embeddings are necessarily unique, as we subsequently show. Hence, we use the *geodesic Laplacian kernel* instead of the Gaussian one because it poses less strict requirements on the induced space, as stated in Proposition 2. Specifically, the metric used in the kernel function needs to be cnd. We cannot directly prove this yet, but we can prove that the converse is not true. To this end, we first notice that the metric space induced by the GWD, which we refer to as $X$, does *not* have a curvature that is bounded from above.

**Definition 8.** *A metric space $(X, d)$ is said to be $\mathrm{CAT}(k)$ if its curvature is bounded by some real number $k > 0$ from above. This can also be seen as a "relaxed" definition, or generalisation, of a Riemannian manifold.*

**Theorem 3.** *$X$ is not in $\mathrm{CAT}(k)$ for any $k > 0$, meaning that its curvature is* not *bounded by any $k > 0$ from above.*

*Proof.* This follows from a similar argument presented by Turner et al. [42]. We briefly sketch the argument. Let $G$ and $G'$ be two graphs. Assume that $X$ is a CAT($k$) space for some $k > 0$. Then, it follows [7, Proposition 2.11, p. 23] that if $D_W^{f_{\text{WL}}}(G, G') < \pi^2/k$, there is a *unique* geodesic between them. However, we can construct a family of graph embeddings for which this is not the case. To this end, let $\epsilon > 0$ and $f_{\text{WL}}(G)$ and $f_{\text{WL}}(G')$ be two graph embeddings with node embeddings $a_1 = (0,0)$, $a_2 = (\epsilon, \epsilon)$ as well as $b_1 = (0, \epsilon)$ and $b_2 = (\epsilon, 0)$, respectively. Because we use the Euclidean distance as a ground distance, there will be two optimal transport plans: the first maps $a_1$ to $b_1$ and $a_2$ to $b_2$, whereas the second maps $a_1$ to $b_2$ and $a_2$ to $b_1$. Hence, we have found two geodesics that connect $G$ and $G'$. Because we may choose $\epsilon$ to be arbitrarily small, the space cannot be CAT($k$) for $k > 0$. $\qquad\square$

Although this does not provide an upper bound on the curvature, we have the following conjecture.

**Conjecture 1.** *$X$ is an Alexandrov space with curvature bounded from below by zero.*

For a proof idea, we refer to Turner et al. [42]; the main argument involves characterizing the distance between triples of graph embeddings. This conjecture is helpful insofar as being a nonnegatively curved Alexandrov space is a necessary prerequisite for $X$ to be a Hilbert space [39]. In turn, Feragen et al. [11] shows that cnd metrics and Hilbert spaces are intricately linked. Thus, we have some hope in obtaining a cnd metric, even though we currently lack a proof. Our empirical results, however, indicate that it is possible to turn the GWD into a cnd metric with proper normalisation. Intuitively, for high-dimensional input spaces, standardisation of input features changes the curvature of the induced space by making it locally (almost) flat.

To support this argumentation, we refer to an existing way to ensure positive definiteness. One can use an alternative to the classical Wasserstein distance denoted as the sliced Wasserstein [32]. The idea is to project high-dimensional distributions into one-dimensional spaces, hereby calculating the Wasserstein distance as a combination of one-dimensional representations. Kolouri et al. [23] showed that each of the one-dimensional Wasserstein distances is conditional negative definite. The kernel on high-dimensional representations is then defined as a combination of the one-dimensional positive definite counterparts.

## A.2 Proof of Lemma 1

*Proof.* We recall the matrix notation introduced in Equation 2 of the main paper, where $M$ is the cost or distance matrix, $P \in \Gamma$ is a transport matrix (or joint probability), and $\langle \cdot, \cdot \rangle$ is the Frobenius dot product. Because we give equal weight (i.e., equal probability mass) to each of the vectors in each set, $\Gamma$ contains all nonnegative $n \times n'$ matrices $P$ with

$$\sum_{i=1}^{n} p_{ij} = \frac{1}{n'} \quad , \quad \sum_{j=1}^{n'} p_{ij} = \frac{1}{n} \quad , \quad p_{ij} \geq 0 \;\; \forall i, j$$

For notation simplicity, let us denote by $D_{\text{Ham}}^h$ the Hamming matrix $D_{\text{Ham}}(f_{\text{WL}}^h(G), f_{\text{WL}}^h(G'))$, where the $ij$-th entry is given by the Hamming distance between the embedding of the $i$-th node of graph $G$ and the embedding of the $j$-th node of graph $G'$ at iteration $h$. Similarly, we define $D_{\text{disc}}^h$ to be the discrete metric distance matrix, where the $ij$-th entry is given by the discrete distance between feature $h$ of node embedding $i$ of graph $G$ and feature $h$ of node embedding $j$ of graph $G'$. It is easy to see that $[D_{\text{Ham}}^h]_{ij} \in [0, 1]$ and $[D_{\text{disc}}^h]_{ij} \in \{0, 1\}$ and, by definition,

$$D_{\text{Ham}}^H = \frac{1}{H} \sum_{h=0}^{H} D_{\text{disc}}^h.$$

Moreover, because of the WL procedure, two labels that are different at iteration $h$ will also be different at iteration $h + 1$. Hence, the following identity holds:

$$\left[ D_{\text{Ham}}^h \right]_{ij} \leq \left[ D_{\text{disc}}^h \right]_{ij},$$

which implies that $[D_{\text{Ham}}^h]_{ij} = 0 \iff [D_{\text{disc}}^h]_{ij} = 0$. An optimal transportation plan $P^h$ for $f_{\text{WL}}^h$ embeddings satisfies

$$\left\langle P^h, D_{\text{Ham}}^h \right\rangle \leq \left\langle P, D_{\text{Ham}}^h \right\rangle \;\; \forall P \in \Gamma.$$

Assuming that $P^h$ is not optimal for $D_d^h$, we can define $P^*$ such that

$$\left\langle P^*, D_{\text{disc}}^h \right\rangle < \left\langle P^h, D_{\text{disc}}^h \right\rangle.$$

Because the entries of $D_{\text{disc}}^h$ are either 0 or 1, we can define the set of indices tuples $\mathcal{H} = \left\{ (i,j) \mid [D_{\text{disc}}^h]_{ij} = 1 \right\}$ and rewrite the inequality as

$$\sum_{i,j \in \mathcal{H}} p_{ij}^* < \sum_{i,j \in \mathcal{H}} p_{ij}^h.$$

Considering the constraints on the entries of $P^*$ and $P^h$, namely $\sum_{i,j} p_{ij}^* = \sum_{i,j} p_{ij}^h = 1$, this implies that, by rearranging the transport map, there is more mass that could be transported at 0 cost. In our formalism,

$$\sum_{i,j \notin \mathcal{H}} p_{ij}^* > \sum_{i,j \notin \mathcal{H}} p_{ij}^h.$$

However, as stated before, entries of $D_d^h$ that are 0 are also 0 in $D_{\text{Ham}}^h$. Therefore, a better transport plan $P^*$ would also be optimal for $D_{\text{Ham}}^h$:

$$\left\langle P^*, D_{\text{Ham}}^h \right\rangle < \left\langle P^h, D_{\text{Ham}}^h \right\rangle,$$

which contradicts the optimality assumption above. Hence, $P^h$ is also optimal for $D_{\text{disc}}^H$. $\qquad\square$

## A.3 Proof of Lemma 2

*Proof.* Intuitively, the transportation plan at iteration $h$ is a "refinement" of the transportation plan at iteration $h - 1$, where only a subset of the optimal transportation plans remains optimal for the new cost matrix $D_H^h$. Using the same notation as for the Proof in Appendix A.2, and considering the WL procedure, two labels that are different at iteration $h$ will also be different at iteration $h + 1$. Hence, the following identities hold:

$$\left[ D_{\text{Ham}}^h \right]_{ij} \leq \left[ D_{\text{Ham}}^{h+1} \right]_{ij} \quad \left[ D_{\text{disc}}^h \right]_{ij} \leq \left[ D_{\text{disc}}^{h+1} \right]_{ij}$$

$$\left[ D_{\text{Ham}}^h \right]_{ij} \leq \left[ D_{\text{disc}}^h \right]_{ij}.$$

An optimal transportation plan $P^h$ for $f_{WL}^h(G)$ embeddings satisfies

$$\left\langle P^h, D_{\text{Ham}}^h \right\rangle \leq \left\langle P, D_{\text{Ham}}^h \right\rangle \ \forall P \in \Gamma,$$

which can also be written as

$$\left\langle P^h, D_{\text{Ham}}^h \right\rangle = \frac{1}{h} \left( (h-1) \cdot \left\langle P^h, D_{\text{Ham}}^{h-1} \right\rangle + \left\langle P^h, D_{\text{disc}}^h \right\rangle \right).$$

The values of $D_{\text{Ham}}^h$ increase in a step-wise fashion for increasing $h$, and their ordering remains constant, except for entries that were 0 at iteration $h - 1$ and became $\frac{1}{h}$ at iteration $h$. Hence, because our metric distance matrices satisfy monotonicity conditions and because $P^h$ is optimal for $D_{\text{disc}}^h$ according to Lemma 1, it follows that

$$\left\langle P^h, D_{\text{Ham}}^{h-1} \right\rangle \leq \left\langle P, D_{\text{Ham}}^{h-1} \right\rangle \ \forall P \in \Gamma.$$

Therefore, $P^h$ is also optimal for $f_{\text{WL}}^{h-1}(G)$ embeddings. $\qquad\square$

## A.4 Proof of Theorem 2

*Proof.* Using the same notation as for the Proof in Appendix A.2 and the formulation in Equation 2, we can write

$$D_W^{f_{\text{WL}}}(G, G') = \min_{P^H \in \Gamma} \left\langle P^H, D_{\text{Ham}}^H \right\rangle$$

$$= \min_{P^H \in \Gamma} \frac{1}{H} \sum_{h=0}^{H} \langle P^H, D_{\text{disc}}^h \rangle.$$

Let $P^*$ be an optimal solution for iteration $H$. Then, from Lemmas 1 and 2, it is also an optimal solution for $D_{\text{disc}}^H$ and for all $h = 0, \ldots, H - 1$. We can rewrite the equation as a sum of optimal transport problems:

$$D_W^{f_{\text{WL}}}(G, G') = \frac{1}{H} \sum_{h=0}^{H} \min_{P^* \in \Gamma} \langle P^*, D_{\text{disc}}^h \rangle. \tag{14}$$

This corresponds to a sum of 1-dimensional optimal transport problems relying on the discrete metric, which were shown to be conditional negative functions [16]. Therefore, the final sum is also conditional negative definite. $\qquad\square$

## A.5 Data sets and additional results

We report additional information on the data sets used in our experimental comparison in Supplementary Table A.1. Our data sets belong to multiple chemoinformatics domains, including small molecules (MUTAG, PTC-MR, NCI1), macromolecules (ENZYMES, PROTEINS, D&D) and chemical compounds (BZR, COX2). We further consider a movie collaboration data set (IMDB, see [47] for a description) and two synthetic data sets SYNTHIE and SYNTHETIC-NEW, created by Morris et al. [27] and Feragen et al. [10], respectively. The BZR-MD and COX2-MD data sets do not have node attributes but contain the atomic distance between each connected atom as an edge weight. We do not consider distances between non-connected nodes [24] and we equip the node with one-hot-encoding categorical attributes representing the atom type, i.e., what is originally intended as a categorical node label. On IMDB-B, IMDB-BINARY was used with the node degree as a (semi-)continuous feature for each node [47]. For all the other data sets, we use the off-the-shelf version provided by Kersting et al. [20].

Results on synthetic data sets are provided in Table A.2. We decided not to include those in the main manuscript because of the severely unstable and unreliable results we obtained. In particular, for both data sets, there is a high variation among the different methods. Furthermore, we experimentally observed that even a slight modification of the node features (e.g., normalisation or scaling of the embedding scheme) resulted in a large change of performances (up to $15\%$). Additionally, it has been previously reported [10, 27] that on SYNTHETIC-NEW, a WL with degree treated as categorical node label outperforms the competitors, suggesting that the continuous attributes are indeed not informative. Therefore, we excluded these data sets from the main manuscript, as we concluded that they could not fairly assess the quality of our methods.

## A.6 Details on hyperparameter selection

The following ranges are used for the hyperparameter selection: the parameter of the SVM $C = \{10^{-3}, \ldots, 10^3\}$ (for continuous attributes) and $C = \{10^{-4}, \ldots, 10^5\}$ (for categorical attributes); the WL number of iterations $h = \{0, \ldots, 7\}$; the $\lambda$ parameter of the WWL $\lambda = \{10^{-4}, \ldots, 10^1\}$. For RBF-WL and VH-C, we use default $\gamma$ parameter for the Gaussian kernel, i.e., $\gamma = 1/m$, where $m$ is the size of node attributes. For the GH kernel, we also fix the $\gamma$ parameter to $1/m$. For HGK, we fix the number of iterations to 20 for each data set, except for SYNTHETICnew where we use

Table A.1: Description of the experimental data sets

| DATA SET | CLASS RATIO | NODE LABELS | NODE ATTRIBUTES | EDGE WEIGHTS | # GRAPHS | CLASSES |
|---|---|---|---|---|---|---|
| MUTAG | 63/125 | ✓ | - | - | 188 | 2 |
| NCI1 | 2053/2057 | ✓ | - | - | 4110 | 2 |
| PTC-MR | 152/192 | ✓ | - | - | 344 | 2 |
| D&D | 487/691 | ✓ | - | - | 1178 | 2 |
| ENZYMES | 100 PER CLASS | ✓ | ✓ | - | 600 | 6 |
| PROTEINS | 450/663 | ✓ | ✓ | - | 1113 | 2 |
| BZR | 86/319 | ✓ | ✓ | - | 405 | 2 |
| COX2 | 102/365 | ✓ | ✓ | - | 467 | 2 |
| SYNTHIE | 100 PER CLASS | - | ✓ | - | 400 | 4 |
| IMDB-BINARY | 500/500 | - | (✓) | - | 1000 | 2 |
| SYNTHETIC-NEW | 150/150 | - | ✓ | - | 300 | 2 |
| BzR-MD | 149/157 | ✓ | - | ✓ | 306 | 2 |
| COX2-MD | 148/155 | ✓ | - | ✓ | 303 | 2 |

Table A.2: Classification accuracies on synthetic graphs with continuous node attributes. Comparison of hash graph kernel (HGK-WL, HGK-SP), GraphHopper kernel (GH), and our method (WWL).

| METHOD | SYNTHIE | SYNTHETIC-NEW |
|---|---|---|
| VH-C | $27.51 \pm 0.00$ | $60.60 \pm 1.60$ |
| RBF-WL | $94.43 \pm 0.55$ | $86.37 \pm 1.37$ |
| HGK-WL | $81.94 \pm 0.40$ | $\mathbf{95.96 \pm 0.25}^*$ |
| HGK-SP | $85.82 \pm 0.28$ | $80.43 \pm 0.71$ |
| GH | $83.73 \pm 0.81$ | $88.83 \pm 1.42$ |
| WWL | $\mathbf{96.04 \pm 0.48}^*$ | $86.77 \pm 0.98$ |

100 (these setups were suggested by the respective authors [10, 27]. Furthermore, because HGK is a randomised method, we compute each kernel matrix 10 times and average the results. When the dimensionality of the continuous attributes $m > 1$, these are normalised to ensure comparability among the different feature scales, in each data set except for BZR and COX2, due to the meaning of the node attributes being location coordinates.

## A.7 Runtime comparison

Overall, we note that WL and WL-OA scale linearly with the number of nodes; therefore, these methods are faster than our approach. Because of the differences in programming language implementations of the different methods, it is hard to provide an accurate runtime comparison. However, we empirically observe that the Wasserstein graph kernels are still competitive, and a kernel matrix can be computed in a median time of 40 s, depending on the size and number of graphs (see Figure A.1). For the continuous attributes, our approach has a runtime comparable to GH. However, although our approach can benefit from a significant speedup (see discussion below and Section 5.2), GH was shown to empirically scale quadratically with the number of graph nodes [10]. The HGK, on the other hand, is considerably slower, given the number of iterations and multiple repetitions while taking into account the randomisation.

To evaluate our approach with respect to the size of the graphs and recalling that computing the Wasserstein distance has complexity $\mathcal{O}(n^3 log(n))$, we simulated a fixed number of graphs with a varying average number of nodes per graph. We generated random node embeddings for 100 graphs, where the number of nodes is taken from a normal distribution centered around the average number of nodes. We then computed the kernel matrix on each set of graphs to compare the runtime of regular Wasserstein with the Sinkhorn regularised optimisation. As shown in Supplementary Figure A.1, the speedup starts to become beneficial at approximately 100 nodes per graph on average, which is larger than the average number of nodes in the benchmark data sets we used.

To ensure good performance when using the Sinkhorn approximation, we evaluate the obtained accuracy of the model. Recalling that the Sinkhorn method solves the following entropic regularisation problem,

$$P^\gamma = \underset{P \in \Gamma(X, X')}{\arg\min} \langle P, M \rangle - \gamma h(P),$$

we further need to select $\gamma$. Therefore, on top of the cross-validation scheme described above, we further cross-validate over the regularisation parameter values of $\gamma \in \{0.01, 0.05, 0.1, 0.2, 0.5, 1, 10\}$ for the ENZYMES data set and obtain an accuracy of $72.08 \pm 0.93$, which remains above the current state of the art. Values of $\gamma$ selected most of the time are $0.3, 0.5,$ and $1$.

## A.8 Performance on isomorphic synthetic graphs

We performed an additional experiment to evaluate the difference between WL and WWL for noisy Erdős–Rényi graphs ($n = 30, p = 0.2$). We report the relative distance between $G$ and its permuted and perturbed variant $G'$, w.r.t. a third independent graph $G''$ for an increasing noise level (i.e., edge removal) in Figure A.2. We see that WWL is more robust against noise.

Figure A.1: Runtime performance of the WWL Kernel computation step with a fixed number of graphs. We also report the time taken to compute the ground distance matrix as `distance_time`. Here, `total_time` is the sum of the time to compute the ground distance and the time taken to solve the optimal transport (ot) problem for the regular solver or the Sinkhorn-regularised one. The logarithmic scale on the right-side figure shows how, for a small average number of nodes, the overhead to run Sinkhorn is higher than the benefits.

Figure A.2: Relative distance between (Erdős–Rényi) graph $G$ and its permuted and perturbed variant $G'$ w.r.t. a third independent graph $G''$ for an increasing noise level.

## Footnotes

[3]We observe that for all considered data sets, after standardisation of the input features before the embedding scheme, GWD matrices are conditional negative definite.