[Reviews · NeurIPS 2019]

Reviewer 1



originality The paper proposes a novel method to compute distances between graphs, that can take into accound the node content as well as the global information repartition, using Wasserstein distance and OT tools, based on existing subtrees kernels. The fact that it can be applied to continuous attributes in nodes is a plus. On that matter, I wonder if links could be done with kernels proposed in [1] (that can also deal with complex information on edges)? quality: The proposed graph kernel is based on known technics thus the theoretical part is not large, it mainly concerns the notion of definitness of the produced kernel. This point is well discussed and proven for the discrete case. For the continuous attributes case, the author have insights but no formal proof. However since they chose to use KSVM as a safety guard in the experimental part, it's not a problem. They also note that results between KSVM and SVM are very similar, probably signifying that either the resulting kernels are indeed positive definite and if not, the negative part if neglectible. clarity: The paper is clearly written and polished. All required tools are described/defined. It's a clear improvement over a previous version I had the oppotunity to review. Associated with the provided code, this work can easily be reproduced and re-used. significance; having efficient and accurate kernels for graph is definitively an important matter. I'm confident that this work is useful both for future research and for practitioners. Remark : there are some arxiv references for papers that are very likely published, please check and update. [1] Kashima et al. Marginalized kernels between labeled graphs, ICML 2003

Reviewer 2



The main motivation of this work is based on the fact that conventional graph kernels loose information in their embedding and/or aggregation steps. While we agree with the authors on this point, it is not clear what is the information lost with the proposed WWL graph kernel. Since the proposed method is based on the WL subtree kernel, then it has the same weaknesses as it. Moreover, it may have more issues, such as the non-uniqueness of the embedding, the iterative operations related to hashing… The part “To ensure the theoretical correctness of our results…” is confusing and misleading. On a first reading, the reader may understand that the theoretical results are not correct. The authors need to rewrite this part in order to emphasize on the fact that the work with indefinite kernel learning and RKKS is only required when working with the continuous WWL. This is not the case of categorical WWL. Moreover, the results given in Table 1 are equivalent to using conventional definite kernels and RKHS. It is difficult to understand the assessment when comparing KSVM (in the case of WWL) and conventional SVM (all other methods). We think that it is important to assess the price to pay when using the proposed indefinite kernel with RKKS. This is relevant because many graph kernels have been proposed with the positive definiteness, thus can be easily used in conventional Machine Learning algorithms. In experiments, it seems that WL-OA and the proposed WWL have comparable performances when dealing with categorical node labels. We think that it would be important to test if a version of WL-OA that takes into account continuous node labels would be also as powerful as the proposed WWL. All experiments would have benefitted from a comparison with more graph kernels from the literature. This would be more important as baseline that using simple vertex or edge histograms. In other words, it is not clear how this more complex approach, which may provide indefinite kernels, compares to simple approaches, i.e., the large class of graph kernels. This is the first time that we read “Reproducible kernel Hilbert space” and “Reproducible kernel Krein space”. It should be “Reproducing”, not “Reproducible” ! -------------------------------------- -------------------------------------- We thank the authors for their positive feedback concerning some of the raised issue. We have updated the overall score accordingly.

Reviewer 3



Comment before rebuttal/discussion: The idea is interesting and potentially useful, though it is not obvious to this reviewer whether the optimal transport metric is the right one in this case. Given to isomorphic graphs, the W-L embedding is not exactly the same vector, but the corresponding permutation will send one into the other one. The Wasserstein distance is actually very sensitive to permutations on the labels so it seems to me that something like the Gromov-Wasserstein distance should be more amenable for the purpose of comparing graphs. The point the authors make in lines 71-80 is that the W-L kernels (that use measurements based on walks and other permutation-invariant features) are too weak and that's why the Wasserstein distance is a better candidate. I may be missing something but I'd like to see a theoretical justification for the Wasserstein distance in this context. Is it true that if two graphs are isomorphic then the WL embeddings are usually essentially the same, and therefore the Wasserstein distance is a relevant metric? Or is the algorithm taking this into consideration in a way I am missing? Comment after rebuttal/discussion: Now I see where my confusion was. I think the notation is a little unusual for W-L. If you look at the outcome of the iterations of W-L as distributions in R^{H+1} (supported in N points where N is the number of vertices in the graphs), then it works (the distributions don't care about the permutation). [Following the W-L algorithm it is intuitive why it works, the sequence of vectors for each node is equivariant with respect to permutations on the original labels]. Before I understood that the authors were computing Wasserstein distances between the different iterations of the W-L algorithm (in that case it'd be distributions in R^N supported in H+1 points). In that case the invariance doesn't hold. The authors may want to clarify this point, and say that they are computing a distance between distributions in R^{H+1}. In particular it may worth mentioning that the distance is 0 whenever two graphs are isometric. Maybe in line 158, right after the statement "While not constituting a test of graph isomorphism..." Originality. This reviewer believes the idea of the paper is interesting. I am not familiar enough with this literature to comment on the originality of the paper. Quality. The paper is very well written and very clear. The only issue I had is what I mentioned in the point above. The rest of the comments were addressed by the response of the authors. Thank you! Feedback on the code: Since I was confused about the embedding in the pair of isomorphic graphs setting, I actually used the code to understand what was going on. The code computes the W-L embedding and then the optimal transport on the embedding. It reads a file with some specific format with many graphs. I think a better interface that should be easy to add is a simple python function that takes two matrices (adjacency of the graph) and computes the embedding and the Wasserstein distance. It's not necessary but it would make it much easier to use. (Actually the implementation is very easy, but I had to look at it in order to understand what the embedding and concatenation of different iterations of W-L was doing). Clarity. The paper is clear except for the point I mention above.

[Author Response · NeurIPS 2019]

We thank the reviewers for their valuable positive feedback and comments. Please find our detailed answers below.

**Reviewer 1**

*Links to other kernels [1] & using edge information:* There is certainly a link to these kernels; the main difference is
that these kernels exploit attribute and structural information by means of random walks (for label sequence generation),
whereas our approach follows a WL subtree-based propagation scheme. Concerning the use of edge information, a
straightforward extension for continuous high-dimensional edge attributes would apply WWL on the *dual* graph (where
each edge is represented as a node, and connectivity is established if two edges in the primal graph share the same
node), then combine it with the kernel on primal graphs via appropriate weighting. We will discuss the link to [1] in our
revision and suggest extensions as future work.

*Bibliography:* Thank you, we checked the references and will update them for the revised version.

**Reviewer 2**

*Information lost with WWL, weaknesses and issues, such as non unique embeddings and hashing:* In the categorical
case our kernel shares the propagation scheme of WL and WL-OA, which can lead to non-unique embeddings in rare
cases. The key difference is that instead of comparing histograms, we compare distributions by means of optimal
transport resulting in more granular similarities and thus higher classification performance. In the continuous case, to
the best of our knowledge, WWL is the first WL-based method that does not rely on hashing; node attribute information
is thus better exploited, as demonstrated by our empirical results.

*Revise usage of KSVM:* Thanks, we will clarify that we only use KSVM for the continuous variant of our kernel.

*Assessment when comparing KSVM and SVM; price to pay for using indefinite kernels:* For non-PSD kernels, KSVM
has been proposed as a suitable replacement of classical SVM in machine learning. In practice, when the kernel is
PSD, KSVM is equivalent to SVM, making the empirical and theoretical comparison of the two methods fair.[1] In our
experiments with continuous attributes, we observed that the WWL kernel matrices are approximately PSD.

*WL-OA with continuous node labels:* WL-OA is based on node label histogram matching, hence it is not straightforward
to extend it to the continuous case. While we agree that this would be interesting, the development of a continuous
WL-OA variant is out of the scope of this work.

*More graph kernel comparisons:* Our experiments run all kernels on the same splits, coupled with a thorough
hyperparameter selection, to guarantee a fair comparison. In an analogous setting, Kriege et al. (2016) showed, in the
categorical case, that the shortest path and graphlet kernels perform worse than WL and WL-OA, so we did not include
them. While we do not think such comparison is essential to our message, we could include it. For the continuous case,
we provide an extensive comparison, including a variety of state-of-the-art kernels designed for continuously attributed
graphs. We will describe the reasoning for our choice of comparison partners in more detail in the revised manuscript.

*It should be "Reproducing" not "Reproducible".* Thanks, we will correct this in the revised paper.

**Reviewer 3**

*Gromov–Wasserstein:* Thanks for the suggestion. Obtaining the
Gromov-Wasserstein distance is computationally more demanding, but
this direction is worth exploring in the future.

*Isomorphism & distance 0:* In the categorical case, for which WL
can be applied, if the two graphs are isomorphic, the embeddings are
guaranteed to be the same and the Wasserstein distance is 0. We are
more interested in calculating the dissimilarity between graphs that
are *not* isomorphic; here, WL and WWL will significantly differ, as
WL is using a linear kernel between the histograms, whereas WWL
characterizes differences in the distribution of labels.

*SVM vs. k-NN & KSVM in WL setting:* We had preliminary results with $k$-NN; following the literature & to ensure a fair
comparison, we used SVM for the final experiments. KSVM is only used when PSD is not guaranteed; it is equivalent
to SVM if the kernel is PSD[1], which is the case for WL, for example.

*Improvements:* Thanks for the interesting suggestion. We performed an additional experiment to evaluate the difference
between WL and WWL for noisy E-R graphs ($n = 30$, $p = 0.2$). We report the relative distance between $G$ and its
permuted and perturbed variant $G'$, w.r.t. a third independent graph $G''$ for an increasing noise level. We see that WWL
is more robust against noise.

## Footnotes

[1]Loosli, Gaëlle, Cheng Soon Ong, and Stéphane Canu. "Technical report: SVM in Krein spaces." (2013).


[Meta-Review · NeurIPS 2019]

The reviewers unanimously liked and recommended to accept the paper. The author feedback and discussion clarified some concerns that the reviewers had initially held.